

# Calibrating and validating the InVEST urban cooling model: Case studies in France and the United States

Perrine Hamel[1,2], Martí Bosch[3], Léa Tardieu[4,5], Aude Lemonsu[6], Cécile de Munck[6], Chris Nootenboom[7],
Vincent Viguié[8], Eric Lonsdorf [8], James A. Douglass[9], Richard S. Sharp[10]

[1]Asian School of the Environment, Nanyang Technological University, 50 Nanyang Avenue, Singapore 639798
[2]Earth Observatory of Singapore, Nanyang Technological University, 50 Nanyang Avenue, Singapore 639798
[3]Urban and Regional Planning Community, École Polytechnique Fédérale de Lausanne, Lausanne, Switzerland
[4]TETIS, INRAE, AgroParisTech, CIRAD, CNRS, Univ Montpellier, Montpellier, France
[5]CIRED, Ecole des Ponts ParisTech, AgroParisTech, Cirad, CNRS, EHESS, Université Paris-Saclay, F-94736, Nogent-sur-Marne, France
[6]CNRM, Université de Toulouse, Météo-France, CNRS, Toulouse, France
[7]Ecognosis, USA
[8]The Department of Environmental Sciences, Emory College, GA, 30322 GA, USA
[9]Natural Capital Project, Dept. of Biology and Woods Institute for the Environment, Stanford University, CA 94305 USA
[10]Spring, 5455 Shafter Avenue, Oakland, CA 94618USA

*Correspondence to*: Perrine Hamel (Perrine.hamel@ntu.edu.sg)

**Abstract.** Understanding the cooling service provided by vegetation in cities is important to inform urban policy and planning. However, the performance of decision-support tools estimating heat mitigation for urban greening strategies is not 20  systematically evaluated. Here, we develop a calibration algorithm and evaluate the performance of the Urban Cooling model developed within the open-source InVEST (Integrated Valuation of Ecosystem Services and Tradeoffs) software. The Urban Cooling model estimates air temperature reduction due to vegetation based on four predictors: shade provision, evapotranspiration, albedo, and building density and was designed for data-rich and data-scarce situations. We apply the calibration algorithm and evaluate the model in two case studies (Paris, France, and Minneapolis-St Paul, USA) by examining 25  the spatial correlation between InVEST predictions and reference temperature data at 1-km horizontal resolution. In both case studies, model performance was high for nighttime air temperatures, which is an important indicator of human wellbeing. After calibration, we found a medium performance for surface temperatures during daytime but a low performance for daytime air temperatures in both case studies, which may be due to model and data limitations. We illustrate the model adequacy for urban planning by testing its ability to simulate a green infrastructure scenario in the Paris case study. The predicted air temperature 30  change compared well with that of an alternative physics-based model ($r^2$=0.55 and $r^2$=0.85, for air daytime and nighttime temperatures, respectively). Finally, we discuss opportunities and challenges for the use of such parsimonious decision-support tools, highlighting their importance to mainstream ecosystem services information in urban planning.



## 1 Introduction

The urban heat island is increasingly documented, varying in European cities from 1 to 10°C with an average of 6°C for a
sample of 110 European cities (Santamouris, 2016). The phenomenon involves an increase in air and surface temperatures in
urban areas due to the modification of the energy budget (Oke, 1982). This has direct health, economic and energy consumption
implications (Lehmann, 2014; Santamouris, 2020) as excessive heat has been associated with increases in energy consumption
for cooling purposes, increases in ground level ozone and particulate matter concentrations, and hospital admissions due to

cardiovascular conditions (Gosling et al., 2009; Hémon & Jougla, 2004; Lai & Cheng, 2009; Reid et al., 2012; Santamouris,
2015; Viguié et al., 2020; Wang et al., 2017). To reduce these effects, policy-makers and urban planners are increasingly
turning to blue-green infrastructure (e.g., street trees, green roofs, urban parks), a cost-effective option for urban cooling that
also produces multiple co-benefits (Bolund & Hunhammar, 1999; Corburn, 2009; Cortinovis & Geneletti, 2019; Rosenzweig
et al., 2006; Villanueva-Solis, 2017).

Blue-green infrastructure influences air temperatures and thermal comfort at several scales. At a local scale – a tree or building,
shading can reduce air temperature under the canopy (Kroeger et al., 2018; McDonald et al., 2016; Shashua-Bar & Hoffman,
2000). Street trees can also indirectly improve pedestrian comfort and reduce the use of air conditioning in neighboring
buildings, thus avoiding additional heat generation (Viguié et al., 2020). Green roofs and walls change the heat and energy
balance of buildings: by absorbing incident solar radiation to support biological functions, vegetation acts as a screen and

reduces seasonal temperature variations – although to a limited extent compared to other insulating materials (Eumorfopoulou
& Kontoleon, 2009). At a larger scale, urban parks provide an "oasis effect" reducing air temperatures by up to 6°C (Eliasson,
1996; Jauregui, 1990; Kroeger et al., 2018; Potchter et al., 2006; Spronken-Smith & Oke, 1999; Yu et al., 2020; Ziter et al.,
2019). The effect is influenced by park size (Cao et al., 2010; Yu et al., 2020), composition (Potchter et al., 2006), and local
climatic conditions (Shashua-Bar & Hoffman, 2000; Yu et al., 2020). In a systematic literature review, Bowler et al. (2010)

showed that parks larger than 2-3 ha are systematically cooler than the rest of the city. Recent work by Wong et al. (2021)
suggests a lower threshold of 1 ha. For such large parks, the cooling capacity depends mainly on evapotranspiration, and can
extend to 800 m, although more frequently closer to 100 m (Kroeger et al., 2018; Wong et al., 2021).

Quantitative estimates of the cooling service provided by vegetation come from two main areas. Research in urban climatology
examines the physical processes contributing to air cooling (Phelan et al., 2015). These studies often involve complex

numerical models –meso-scale or micro-scale (Lemonsu et al., 2012; Meili et al., 2020), which focus on one or several
processes explaining microclimate and increasingly incorporating vegetation effects (Bartesaghi Koc et al., 2018). Such
models require long development and calibration processes and therefore significant human and time resources. The same
applies to geostatistical models that require large geospatial datasets (e.g., Cheung et al., 2021). On the other hand, an
increasing body of literature in the fields of urban ecology and ecosystem services science examines the cooling service through

indicators associated with land use (Derkzen et al., 2015; Farrugia et al., 2013; Larondelle & Haase, 2013; Nedkov et al., 2017;
Zardo et al., 2017), as reviewed in previous work (Hamel et al., 2021). These models further simplify the physical processes





underlying the cooling dynamics, the effect of the morphological parameters of the buildings, and the daily or seasonal temperature variations, often by attributing a "cooling capacity" to a broadly defined land use type.

Cooling service models routinely measure the cooling effect of vegetation with land surface temperature, which is readily

accessible from satellite data (Manoli et al., 2019; Zhao et al., 2014). Surface temperature influences thermal comfort and is often correlated with air temperature, making it an interesting proxy for microclimate studies. However, this temperature alone does not suffice to estimate economic or health implications of the urban heat island (Martilli et al., 2020; Venter et al., 2021), reducing the relevance of models estimating land surface temperature in policy making.

These model limitations lead to a lack of decision-support tools that accurately quantify the cooling effect of vegetation on air

temperatures and therefore its impact in socio-economic terms. A recent review of open-source tools found that only a few ecosystem services software tools were adequate for general urban planning purposes—i.e. applicable in any geography, and flexible enough to represent different types of blue-green infrastructure and decision contexts (Hamel et al., 2021). InVEST (Integrated Valuation of ecosystem services and tradeoffs) is one of the most popular tools developed to support ecosystem services assessments by quantifying and mapping the benefits provided by blue-green infrastructure in urban or non-urban

environments (Natural Capital Project, 2022). The Urban Cooling model within InVEST, developed by the author team, addresses some of the limitations stated above by estimating air temperature (instead of land surface temperature only), being applicable in cities all around the world by using readily available data.

The InVEST Urban Cooling model has recently been applied to a European urban dataset (Cortinovis et al., 2022), and to individual cities in Europe, and India (Bosch et al., 2021; Kadaverugu et al., 2021; Zawadzka et al., 2021). However, attempts

to validate the model against observed or modelled data are much rarer. In previous work, Bosch et al. (2021) found good performance of the model in Lausanne, although the observation data was sparse with 11 weather stations over 112 km$^2$. Zawadzka et al. (2021) found that the model explained between 48% and 60% of the variability in land surface temperature for a Summer day in three towns in the United Kingdom. Despite these notable efforts, the model has not been extensively tested and validated against spatialized air temperature data.

The goal of this paper is to evaluate the performance of the InVEST Urban Cooling model in different contexts: daytime and nighttime, and with different levels of data availability. In doing so, we further develop an open-source calibration algorithm to facilitate future applications of the model (Bosch et al., 2021). Given the intended use of InVEST as a decision-support tool, we focus model performance assessment on two aspects: the ability to represent spatial variations in temperatures, and the ability to represent temporal changes in temperature due to landscape changes. We test the model through two case studies

with contrasted climates: the Paris metropolitan area, France, and the Twin Cities, Minnesota, USA. In the following, we present the model performance assessment and discuss the strengths and limitations of the tool and its recommended use to inform urban planning policies.



## 2 Methods

We assess the performance of the InVEST model for the current land uses land covers (LULCs) by comparing its outputs with

available ("best estimates") air temperature and land surface temperature. Since daytime temperatures are highly influenced by convection and atmospheric turbulence, confounding the effect of land use (Le Roy et al., 2020), we hypothesized that the InVEST daytime outputs would better capture variations in land surface temperatures, rather than air temperatures. On the contrary, nighttime temperatures are strongly influenced by land cover (in particular built infrastructure) and thermal processes, so we expected to find a stronger correlation between InVEST and nighttime air temperature. Given the purpose of the model

to support decision-making, we also assess the model's ability to capture the effect of a change in LULC on urban cooling.

### 2.1 Case studies

#### 2.1.1 Paris metropolitan area

The Paris metropolitan area, situated in the Île-de-France administrative region, is our main case study. Île-de-France spans over 12,000 km$^2$, with land use being predominantly agriculture (50%), forests (24%), and artificial areas (22.5%) (see Figure

A1 in Appendix). The climate in Paris is oceanic (humid temperate, no dry season and warm summers). Average rainfall over the period 1981-2010 was 637 mm and summer (June to August) temperatures averaged 19.7°C, with increasingly frequent temperature peaks in the middle of summer (Météo-France). The study period is the heat wave of 8-13 August 2003, described in previous work (de Munck et al., 2018), which led to maximum daytime temperatures of over 39°C in the region. A greening scenario for the Paris metropolitan area was developed by de Munck et al. (2018) to simulate the implementation of low and

high vegetation (60% grass and small shrubs 40% of deciduous trees) for 50% of available ground surface (except roads and areas already covered by vegetation). This represents the greening of pavements, squares, carparks, and some roofs over a surface of 199 km$^2$, or a 23% increase in green areas, resulting in a reduction of up to 2°C in maximum daily mean temperatures over the study area (de Munck et al., 2018).

#### 2.1.2 Twin Cities

The second case study, to validate the model in a different climate zone, is in the Twin Cities metropolitan area surrounding the cities of Minneapolis and St Paul in Minnesota, USA (see Figure A2 in Appendix). The climate of the Twin Cities is classified as hot-summer humid continental. We studied the regional heat wave event of July 22, 2016 and average air temperatures over the period 2011-2014, building on previous studies of urban heat island in the region (Smoliak et al., 2015).

### 2.2 Model description

The InVEST Urban Cooling model, hereafter "InVEST model", is fully described in the InVEST software User's manual (Natural Capital Project, 2022). We provide a summary of key equations here to orient readers. The model computes air temperature prior to air mixing ($T_{nomix}$, in degree Celsius) on each pixel as a function of a background rural reference





temperature $(T_{ref})$ modified by a local heat factor. The latter is expressed as the maximum urban heat island intensity for the city $(UHI_{max})$ modulated by the local heat mitigation $(HM)$, such that, for a given pixel i:

$$T_{nomix}(i) = T_{ref} + \left(1 - HM(i)\right) \times UHI_{max}$$


[1]

While this temperature does not account for mixing due to atmospheric turbulence, actual air temperature, $T_{air}$, is estimated from $T_{nomix}$ using a spatial moving average algorithm with search radius r$_{mix}$. This radius varies with time, depending on the lateral mixing due to atmospheric turbulence, and can be estimated through calibration (Section 2.4).

The proportion of heat mitigation relative to $UHI_{max}$, with values ranging between 0 and 1, is derived from the cooling capacity (CC) of the LULC type on a given pixel and the proximity to large parks. Following an approach proposed by others (Kunapo et al., 2018; Zardo et al., 2017), the cooling capacity during daytime is expressed as a function of shade, evapotranspiration, and albedo:

$$CC(i) = W_S.S(i) + W_A.A(i) + W_E.E(i),$$


[2]

where S(i), A(i), and E(i) are unitless indices ranging from 0 to 1 that characterize shade, albedo, and evapotranspiration on the pixel i, respectively. They are each weighted by a coefficient (W$_S$, W$_A$, W$_E$) constant across the study area. S represents the proportion of shade for a given LULC type, e.g.,1 for a land cover type completely covered by canopy or high buildings, and 0 for bare lands. A is the albedo value of the land cover. E is calculated from reference evapotranspiration (a raster termed

$ET_0$, in mm) and the crop coefficient ($K_c$, no unit) for the LULC type, and then normalized by the maximum value of reference evapotranspiration in the area of interest according to:

$$E(i) = \frac{K_c(i).ET_0(i)}{max(ET_0)}$$

[3]

For nighttime temperatures, the model calculates cooling capacity as the complement of building density (B), an indicator

representing urban compactness and highly correlated to heat storage capacity (Wong et al., 2021). Building capacity is normalized across the area and can be obtained from building density data: a value of 1 implies that the pixel is covered by the buildings with the highest energy retention, often the tallest building, corresponding to a maximum nighttime urban heat island effect:

$$CC(i) = 1 - B(i),$$


[4]

To account for the effect of large parks (>2 ha), both in nighttime and daytime, the heat mitigation factor $HM$ equals a distance-weighted average of the CC values from surrounding areas $(CC_{park})$. The algorithm for this distance-weighted average, for a pixel i, is as follows:

$$GA(i) = area \cdot \sum_{j \in c(d_{cool}^\circ)} g(j)$$




$$CC_{park}(i) = \sum_{j \in c(i)} g(j) \cdot CC_j \cdot exp(-d(i,j)/d_{cool})$$

[5a]

[5b]

$$HM_i = \begin{cases} CC_i & if \ CC_i \geq CC_{park_i} \vee GA_i < 2ha \\ CC_{park_i} & otherwise \end{cases}$$

[5c]

where GA(i) is the total area of green spaces in a buffer of radius $d_{cool}$ (the distance over which a green space has a cooling effect); $area$ is the area of a pixel in ha; c($d_{cool}$) is the buffer area of radius $d_{cool}$; $g(j)$ is 1 if pixel j is green space, 0 otherwise; $CC_{park}(i)$ is the cooling capacity including the influence of parks; and $d(i,j)$ is the distance between pixel i and j.

In plain words, if the amount of green spaces surrounding a pixel (GA) is less than 2 ha, the value of $HM$ on the pixel equals CC (Eq. 5c), assuming little cooling effect outside the park other than through air mixing due to atmospheric turbulence

(defined by $r_{mix}$). The threshold size of 2 ha is obtained from the literature (Bowler et al., 2010; also see Discussion).

## 2.3 Input and calibration data

### 2.3.1 Paris metropolitan area

*Input data.* LULC data for the Île-de-France region were obtained from the regional urban planning agency (Institut Paris Region, 2019) for the year 2003 (Figure S1). Reference evapotranspiration ($ET_0$) for August was obtained from monthly

modeled climatological data for the region, averaged for the 1985-2005 period (ALADIN model, Stéfanon et al., 2015). LULC parameter values for shade, crop evapotranspiration coefficient, albedo, and building intensity were assigned based on a combination of expert opinion and literature review (see Appendix B).

*Reference temperature data.* In the Paris case study, we compared the InVEST model outputs with two datasets of land surface temperature (LST) and air temperature. LST maps with a 1-km horizontal resolution were retrieved from MODIS satellite

products (Wan, 2013). They were obtained for August 13[th] 2003 at daytime and nighttime since data were missing at the two dates closer to the peak of the heat wave (August 11[th] and 12[th]). We also considered higher resolution data from Landsat but due to its lower frequency (16 days) there was no data around the study period. MODIS LST data have been used in many surface UHI studies globally and regionally (Chakraborty & Lee, 2019; Li et al., 2017).

Predictions of air temperature maps with a 1-km horizontal resolution were obtained from the simulations performed by de

Munck et al. (2018) with the physical land-surface model TEB/Surfex (Lemonsu et al., 2012; Masson, 2000; Masson et al., 2013). The model computes the energy and water budgets on a geographic domain of 100 km x 100 km centered on Paris with a user-defined regular grid, and for natural and urban areas, taking land cover and building characteristics as inputs (Masson, 2000). The 1[st] and 99[th] percentiles were extracted from these air temperature data to define the extreme temperature conditions over the domain during the heat wave (to exclude visible outliers). The background reference rural temperature ($T_{ref}$) was set

to the 1[st] percentile, while the maximum urban heat island intensity ($UHI_{max}$) was the difference between both percentiles.





Because the TEB/Surfex data were available for a 100 km x 100 km window, the rest of the analyses are presented for this domain, rather than the entire Ile-de-France region. The three datasets InVEST, TEB/Surfex and MODIS were resampled to a common 1-km grid for comparison, using bilinear interpolation.

### 2.3.2 Twin Cities

*Input data.* We used 2016 LULC data from the National Land Cover Dataset (NLCD; Homer et al., 2020). Details on LULC parameters values are provided in Appendix B and in previous work (Hamel et al., 2021). Reference evapotranspiration for July 2016 was obtained from globally available data from the Consultative Group on International Agricultural Research (Trabucco & Zomer, 2018).

*Reference temperature data.* We compared air temperature as modeled by InVEST to two data sources: the midday LST maps
of the study area during a regional heatwave on July 22, 2016, retrieved from Landsat data at a 30-m horizontal resolution for daytime and MODIS for nighttime (Wan, 2013); and rasters of average summertime (June, July, August, 2011-2014) daytime and nighttime air temperatures across the study area, interpolated from a dense network of temperature sensors (Smoliak et al., 2015). We projected the reference temperature data to match the InVEST output coordinate system and resampled all datasets to match the resolution (~1 km) of Smoliak et al.'s data using bilinear interpolation. The background reference rural
temperature was defined using the average air temperature in July for the metropolitan area obtained from the US National Weather Service (23.2 °C; NOAA 2020) and the maximum urban heat island intensity was taken from a global assessment of urban heat islands (2.05°C; Chakraborty & Lee, 2019).

### 2.4 Model calibration and performance assessment

For calibration, we further developed an optimization algorithm used in previous work (Bosch et al., 2021). The algorithm
starts with default parameter values and implements a simulated annealing optimization to derive the values of the five parameters for daytime ($r_{mix}$, $d_{cool}$, $W_a$, $W_e$, $W_s$) or two parameters for nighttime ($r_{mix}$, $d_{cool}$,) until the algorithm converges to a solution that minimizes $r^2$ (or a limit number of 100 iterations is reached). The source code for the calibration tool can be found at: https://github.com/martibosch/invest-ucm-calibration/tree/v0.6.0 (and on Zenodo: Please refer to the Code Availability section) and a user guide available at: https://invest-ucm-calibration.readthedocs.io/en/latest/user-
guide.html. In addition, we performed a one-at-a-time sensitivity analyses to further understand how the calibration parameters influence model outputs. Ranges of variation for each parameter are provided in Appendix C.

To demonstrate the application of the InVEST model in practice and assess the model's ability to represent the effect of a change in LULC, we also examined the effect of a greening strategy on urban cooling in the first case study, where comparison data were available from an alternative model. We used the greening scenario LHV50 (50% of impervious areas covered by
low and high vegetation) developed and simulated by de Munck et al. (2018) and described in Section 2.1. In InVEST, we represented this scenario by changing the LULC properties (shade, albedo, crop coefficient) for urban categories. Specifically,



we estimated the proportion of available ground, as defined above, for each category of the LULC map, and computed the weighted average of initial parameter values and values for an urban forest. For example, for the "Parks or Gardens" LULC category, it was estimated that 15% of the ground was available, so a weighted average of the shade value for Forest (1) and original park value (0.5) was computed. The resulting parameter values are shown in Supplementary Material.

## 3 Results

### 3.1 Model performance prior to calibration

#### 3.1.1 Daytime temperatures

For the Paris case study, prior to calibration, the InVEST daytime temperatures were moderately correlated with daytime LST ($r^2$=0.60), with a mean absolute error (MAE) of 3°C, but showed no correlation with air temperatures (Table 1). In the Twin Cities, the InVEST results had low correlation with LST ($r^2$=0.20) but slightly higher for air temperatures ($r^2$=0.29). Of note, comparisons between InVEST and LST in the Twin Cities revealed a large MAE (11.90°C) as surface temperatures can be much higher (up to 52°C) than air temperatures (up to 26°C) based on the reference temperature data.

Visual observations of the difference between the two maps suggest that the model overestimates the temperatures in forested areas (e.g., to the west and southern areas, see Figure S1 for LULC map), as well as in the dense urban areas in the city of Paris (centre, Figure 1). In the Twin Cities, the model overestimates temperatures in agricultural areas, while underestimating temperatures around wetland areas (Figure 2 and Figure S1 for detailed LULC map). In both case studies, the model exhibited a higher variability than reference data.

#### 3.1.2 Nighttime temperatures

As expected, the InVEST nighttime temperatures were correlated with both air and surface temperature, although more strongly with air temperature in Paris ($r^2$=0.84 for Paris, and $r^2$=0.70 for the Twin Cities, Table 1). The MAE is also significantly lower when calculated with respect to air temperatures than land surface temperatures for both cases (0.5 against 3.0°C for Paris and 0.5 against 2.5°C for Twin Cities).

Nevertheless, the maps of the differences between InVEST and nighttime air temperature show important spatial heterogeneities. For Paris, this suggests that the model severely underestimates temperatures in the core area (by about 3°C) and overestimates temperatures in the new urban developments (by about 1°C, Figure 1 and Figure S1). In the Twin Cities, InVEST systematically underestimates temperatures in the city center, outlying developed suburbs, and around bodies of water (e.g. in the west and the confluence of the Minnesota and Mississippi rivers south of the city center), while overestimating temperatures in the surrounding agricultural hinterlands (Figure 2 and Figures S2).





## 3.2 Model performance after calibration

The model performance remained relatively stable after calibration for nighttime air temperatures in both Paris and the Twin Cities, reaching $r^2$ values of 0.84 and 0.73, respectively. Daytime temperatures, however, showed very low correlations with best estimates in Paris and medium correlation ($r^2$=0.33) in the Twin Cities. The sensitivity analysis conducted for the Paris case study confirmed that most model parameters had only a limited effect on correlations with LST data (Figures C1 and C2 in Appendix C). Only $r_{mix}$ had a significant effect on $r^2$ (with a range of $r^2$ values ranging from 0.30 to 0.62 for $r_{mix}$ ranging from 100 to 5000 m).

The Twin Cities daytime calibrated values of the parameters are reported in Table 2 (with UHI$_{max}$=11.7°C). The shade weight is the highest after calibration. Nighttime calibrated parameter values suggest that there is less air mixing, with both $r_{mix}$ and $d_{cool}$ having lower values than during daytime in the Twin Cities (Table 2).

**Table 1: InVEST model performance for daytime and nighttime air (Tair) and surface (LST) temperatures (MAE: mean absolute error) before and after calibration. Post-calibration values are shown in parentheses.**

|              |       | Paris | | Twin Cities | |
|--------------|-------|-------------|-----------|-------------|-----------|
|              |       | $r^2$ | MAE (°C) | $r^2$ | MAE (°C) |
| T$_{air}$    | Day   | <0.01 (0.01) | 1.40 (1.30) | 0.29 (0.33) | 0.48 (0.43) |
|              | Night | 0.84 (0.84) | 0.52 (0.52) | 0.70 (0.73) | 0.48 (0.48) |
| LST          | Day   | 0.60 (0.62) | 3.06 (2.80) | 0.20 (0.43) | 11.9 (8.70) |
|              | Night | 0.45 (0.47) | 2.99 (6.10) | 0.59 (0.76) | 2.54 (2.48) |

**Table 2: Calibrated coefficient values for air temperature in the two case studies.**

|           |                      | Paris | Twin Cities |
|-----------|----------------------|-------|-------------|
| Daytime   | $r_{mix}$ (m)        | n.a.  | 771         |
|           | $d_{cool}$ (m)       | n.a.  | 109         |
|           | $W_a$ ; $W_e$ ; $W_s$ | n.a.  | 0.21; 0.17; 0.62 |
| Nighttime | $r_{mix}$ (m)        | 500   | 630         |
|           | $d_{cool}$ (m)       | 100   | 66          |





**Figure 1: Difference between modelled and reference air temperatures (°C) for the Paris region for daytime and nighttime simulations, pre- and post-calibration**





**Figure 2: Difference between modelled and reference air temperatures (°C) for the Twin Cities region for daytime and nighttime simulations, pre- and post-calibration**

## 3.3 Effect of the greening scenario

Given the limited effect of calibration for daytime and nighttime air temperature data (see Section 3.2), we tested the correlation obtained with temperature estimates from the uncalibrated model. The results were satisfactory, with a medium correlation strength between InVEST and the TEB/Surfex model ($r^2$= 0.55 and mean absolute error of 0.07 °C) for daytime data (Figure 3). For nighttime data, correlation between the two models was stronger, with $r^2$= 0.85.



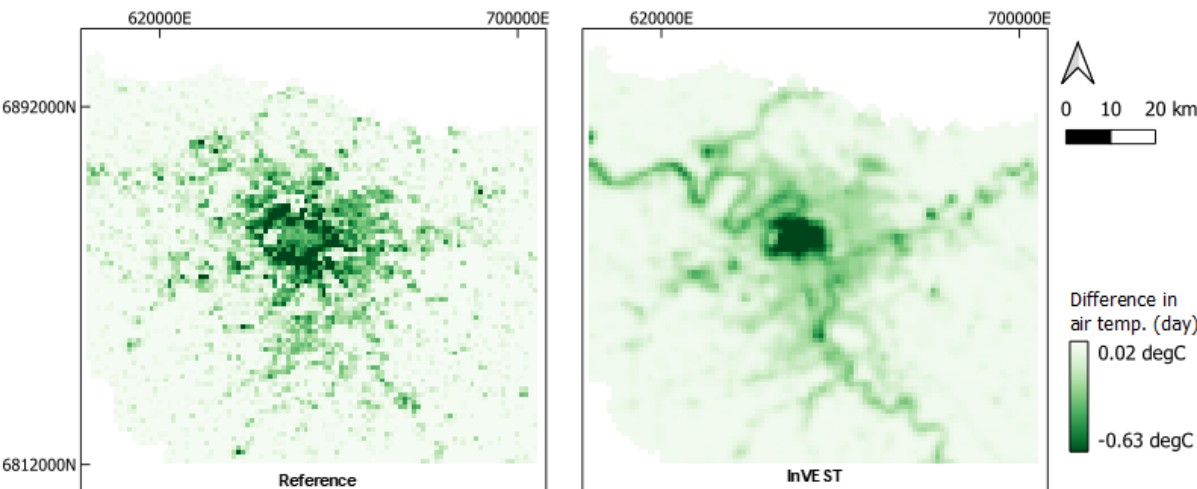

*Figure*

**Figure 3: Cooling service (°C) during daytime resulting from the greening scenario as simulated by InVEST and the TEB/Surfex model (reference data).**

## 4 Discussion

### 4.1 Model calibration and performance

Based on the two case studies, the InVEST model performance was best ($r^2$>0.73) for nighttime air temperature after calibration. The performance improvement due to calibration was modest in most cases (day or night, air or surface temperature), especially for nighttime air temperature in Paris where the performance was already high prior to calibration in Paris ($r^2$=0.84). Only in the Twin Cities for surface temperature did the calibration significantly improve model performance. The modest effect due to calibration can be explained by a relatively low sensitivity to model parameters, which is suggested by the sensitivity analyses presented in Appendix (Figures C3 and C4).

For the Twin Cities, nighttime calibrated parameter values ($r_{mix}$ and $d_{cool}$) are lower than for daytime, which supports the lower convection and air mixing during the night. Calibrated values of $r_{mix}$ are 771 m and 660 m for daytime and nighttime, respectively, which compares well with the estimate of 600 m obtained from previous studies (Lonsdorf et al., 2021; Schatz & Kucharik, 2014). Similar reasoning to verify the physical interpretation of model parameters could be done for the shade, albedo, and evapotranspiration weights if such data are available in a case study: for example, calibrated values could be compared with relative proportion of shading to evapotranspiration that are studied in some cities (e.g., in Singapore, Tan et al., 2018). In both case studies, model performance for surface temperatures ($r^2$ ranging from 0.43 to 0.76 post-calibration) compares with the study by Zawadzka et al. (2021), who found that the InVEST model explained between 48% and 60% of the variability in land surface temperature for a Summer day in three towns in the United Kingdom.



Regarding scenario assessment, the model in the Paris case study performed reasonably well ($r^2$= 0.55 and $r^2$= 0.85 for air
daytime and nighttime temperatures, respectively). This confirms the potential of the tool to support urban planning studies
where different scenarios might need to be compared based on their cooling potential, similar to earlier work with InVEST
(Bosch et al., 2022). The effect from the greening scenario, resulting in less than 1°C cooling, is modest but in line with the
literature on such large-scale implementation of green infrastructure, not only from the reference study by de Munck et al.
(2018), but also for other temperate climate studies.

### 4.2 Limitations of the study and future works

In both our case studies, we note that the performance was poor for daytime air temperature, which does not support the use
of the model for absolute temperature estimates. This poor performance may be due to oversimplifications of the physical
processes involved in urban cooling. In particular, the simplification of the flow dynamics means that a model such as InVEST
cannot represent urban canyon and wind effects in the city. The effect of parks is also simplified with a threshold of 2 ha for
park size. While this has the advantage of reducing the number of model parameters, it also ignores the effect of smaller
greenspaces might have on their surroundings (Wong et al., 2021; Yu et al., 2020).

An important limitation of any urban climate study examining fine-resolution spatial variations in temperature is the
availability of robust reference data. Because air temperature cannot be readily derived from remote sensing data, models are
routinely compared to networks of weather stations (Bosch et al., 2021; de Munck et al., 2018; Smoliak et al., 2015). This
means that model performance is only assessed for a limited number of points, which are not typically representative of the
diversity of LULC in a region. When data are collected specifically for model validation purposes (e.g., transect data), they
are also limited by practical factors such as timing considerations (e.g., mismatch in time between the beginning and end of
the transect) or lack of reference data (Stewart, 2011; Velasco, 2018).

In our study, we have used alternative models as reference data for air temperatures, either a physics-based model or a statistical
interpolation model. Both of these models have limitations and uncertainties in themselves, making a fine scale understanding
of the limitations of the InVEST model challenging. In other words, differences in models observed in Figures 1 and 2 might
also be due to errors in the reference data and future studies could examine the effect of uncertainty in reference datasets on
calibration and model performance. LST datasets, on the other hand, are less prone to such limitations and are more robust
when it comes to spatial distributions. Absolute values are nonetheless challenging to ascertain requiring correction algorithms.
Finally, an important study limitation that could be explored in future work is the spatial resolution and scale of case studies.
In both cases, the InVEST model was evaluated at 1-km horizontal resolution over an area of about 100 km by 100 km. Because
the model produces outputs at the same resolution as LULC inputs, it would be interesting to evaluate its performance at a
different scale and resolution, where such data are available. Relatedly, the use of local climate zones, which is common in
urban climate studies (Aslam & Rana, 2022), could be examined as an alternative parameterization of the InVEST model. This
work would also examine the influence of reference temperature ($T_{ref}$) and maximum intensity ($UHI_{max}$), as well as the
biophysical parameter values, which have not been explored in the present study.



Overall, this discussion of data quality and calibrated parameter values highlights an important contribution of this study, in the customized calibration tool available on Github (https://github.com/martibosch/invest-ucm-calibration and on Zenodo, please refer to the Code Availability section). Such a tool can be applied to any other city where the InVEST Urban Cooling model is applied with the only data requirement, beyond the InVEST model input data, being a reference temperature dataset (either point data or raster). The calibration tool allows for systematic calibration and model testing, which paves the way for better understanding of model limitations and strengths. Although in our case studies the performance improvement from calibration was modest, future work could assess the performance of the model over multiple cities with comparable datasets, examine the potential of local climate zones for improved parameterization, or explore finer temporal resolution by linking the night and day model outputs.

## 5 Conclusion

In this study, we have developed a custom calibration tool to assess the performance of the InVEST Urban cooling model in two case studies: in Paris, France, and the Twin Cities, MN, USA. Our analyses expand on past model testing studies by providing a much more extensive validation dataset of air temperatures (continuous data based on a reference urban climate model in Paris, and on data interpolated from a dense weather station network in the Twin Cities). The model showed good performance, assessed through mean absolute error (0.52°C) and $r^2$ (0.84) for nighttime air temperatures. Calibration only slightly improved model performance in the Twin Cities. For the case study of Paris, the use of the tool for scenario assessment was supported by moderate (daytime) and high (nighttime) correlation with change predicted by an alternative, physics-based model ($r^2$=0.55 and $r^2$=0.85, for air daytime and nighttime temperatures, respectively). With respect to the study objectives, we conclude that the open-source model can be used to support decisions related to land use and land cover change in cities, with greater reliability for nighttime UHI applications and for relative change (i.e., comparing scenarios to one another as opposed to using absolute values of model predictions. As these results were obtained for the case studies of Paris and the Twin Cities, the InVEST model and calibration tools should be tested in other geographies to assess model performance for urban planning applications. For research applications, future studies in other geographies will help further understand the effect of data resolution and data quality on model performance.





# Appendices

## A Input LULC data

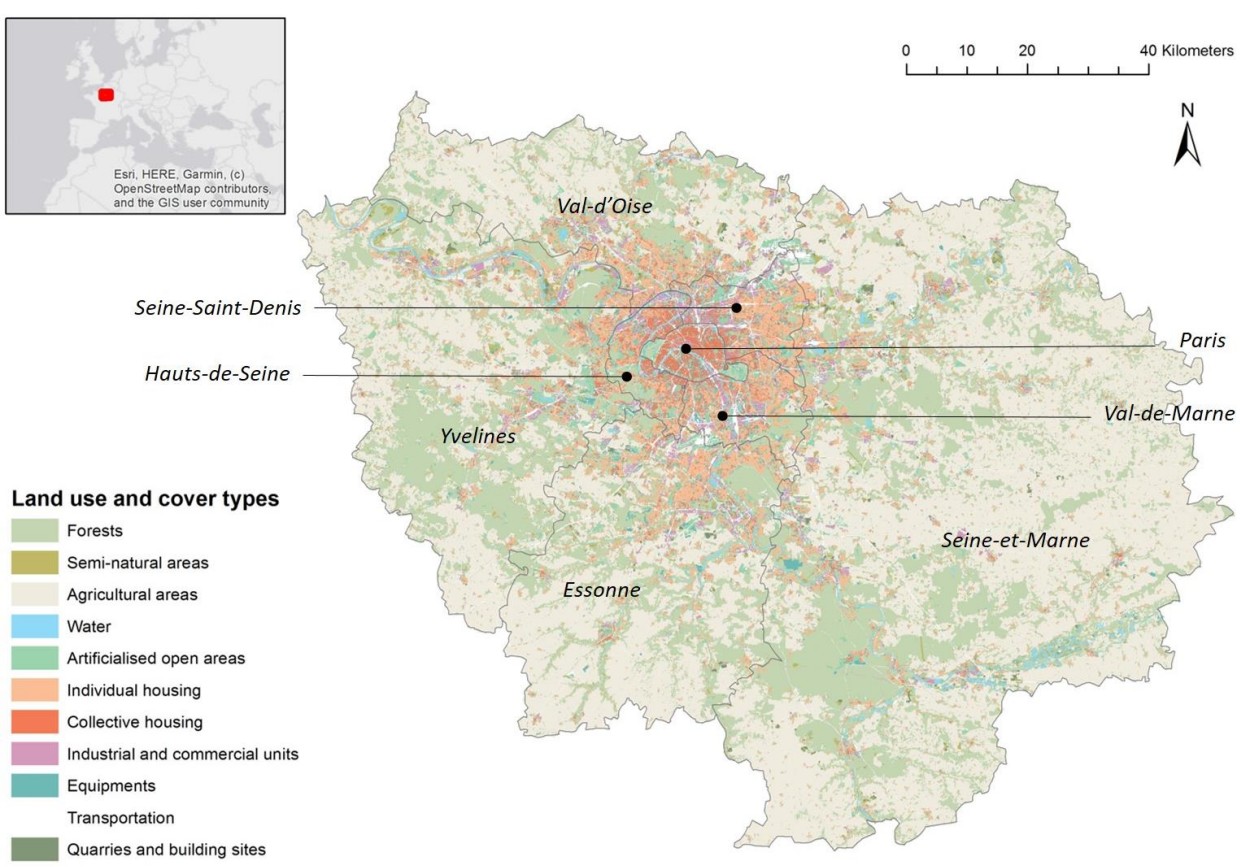

**Figure A1: Land use and cover in Ile-de-France region in 2017, France (based on data developed by the Institut Paris Region)**





**Figure A2: Location and land cover of the Twin Cities metropolitan area in Minnesota, USA, with the cities of Minneapolis and St. Paul outlined.**




**B InVEST biophysical tables**

InVEST biophysical tables can be found in Supplementary Material.

Values for the Paris biophysical table were derived from the following three sources:

- *APUR. (2020). Atténuer les îlots de chaleur urbains - Cahier n°5 : méthodes et outils de conception des projets.*
*https://www.apur.org/fr/nos-travaux/attenuer-ilots-chaleur-urbains-cahier-5-methodes-outils-conception-projets (Accessed 30 April 2023)*

- *Lavigne, P., P. Brejon, and Fernandez. 1994. Architecture Climatique: Une Contribution Au Développement Durable. Edisud.*

- *Stewart, I.D., Oke, T.R., (2012). Local Climate Zones for Urban Temperature Studies. Bulletin of the American*
*Meteorological Society 93, 1879-1900*

Details on the Twin Cities biophysical tables are provided in Supplementary Information of this source:

- *Hamel, P., Guerry, A. D., Polasky, S., Han, B., Douglass, J. A., Hamann, M., Janke, B., Kuiper, J. J., Levrel, H., Liu, H., Lonsdorf, E., McDonald, R. I., Nootenboom, C., Ouyang, Z., Remme, R. P., Sharp, R. P., Tardieu, L., Viguié, V., Xu, D., ... Daily, G. C. (2021). Mapping the benefits of nature in cities with the InVEST software. Npj Urban*
*Sustainability, 1(1), 25. https://doi.org/10.1038/s42949-021-00027-9*

**C Sensitivity analyses**

For the Paris case study, we performed a local sensitivity analysis for the five following parameters (ranges for each parameter are in parentheses):

- Air mixing distance $r_{mix}$ (50 to 5000m)
- Green area maximum cooling distance $d_{cool}$ (50 to 1000m)
- Cooling capacity factors $W_s$, $W_a$, and $W_e$ (0 to 1)

We assessed results based on the correlation with daytime and nighttime temperatures and land surface temperatures. We found that the model was highly sensitive to $r_{mix}$ (Figure S1), with a local maximum for a value of 500 m for daytime temperatures and 1000 m for nighttime temperatures (Figure S3). The model was less sensitive to the values of maximum
cooling distance and the cooling capacity factors: $r^2$ varied between 0.61 and 0.63 for $d_{cool}$, and 0.59 et 0.64 for the weight factors.

These results held for nighttime temperature, where InVEST was also most sensitive to the air mixing parameter, rmix (Figure S4). Sensitivity to the building density parameter B and the maximum cooling distance dcool was much lower.






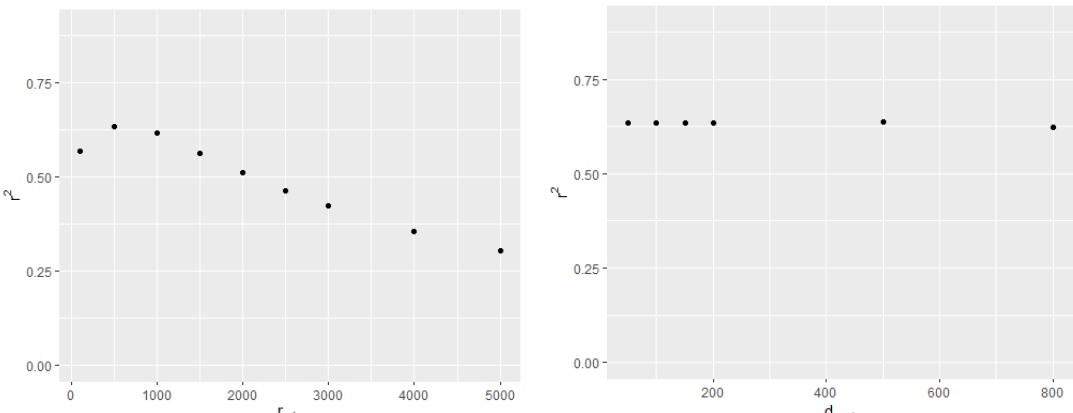

**Figure C1: Sensitivity of the InVEST model to the rmix and dcool parameters and We and Ws factors. Results are plotted against the correlation coefficient ($r^2$) with daytime land surface temperature (MODIS).**

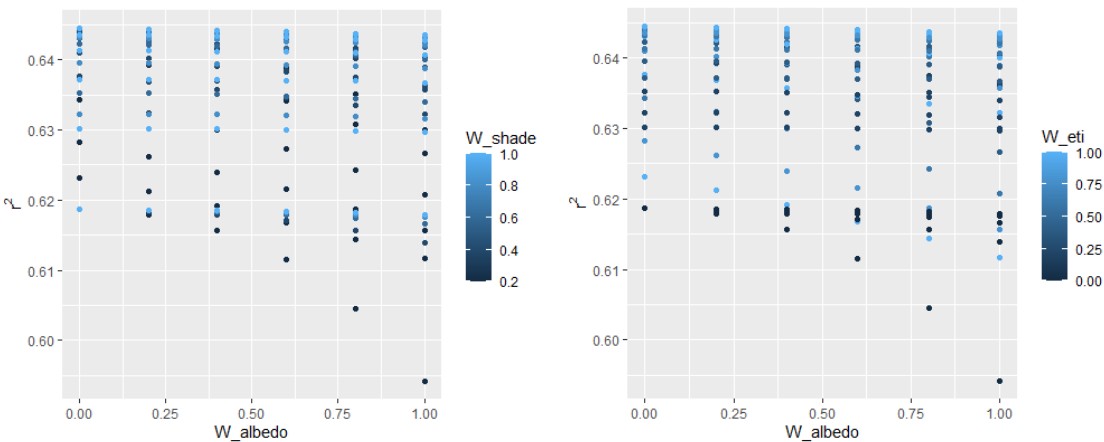


**Figure C2: Sensitivity of the InVEST model to daytime weight factors $W_e$ (W_eti), $W_a$ (W_albedo) and $W_s$ (W_shade). Results are plotted against the correlation coefficient ($r^2$) with daytime land surface temperature (MODIS).**

**Code availability**

The code materials used in this article are available at: https://zenodo.org/record/8081822 (last access: 30 April 2023).

**Data availability**

The source files for the biophysical tables to run the InVEST models can be found in Supplementary Information.

Other input data such as the LULC map for the Paris case study are not publicly available.



**Author contributions**

PH designed the study and conducted the analysis with MB and CN, with technical input from LT, AL, CdM, and VV. JAD and RPS led the software development for the InVEST Urban Cooling model and MB developed the code for the calibration algorithm. PH wrote the first draft of the paper with inputs from LT and CN. All authors provided comments and contributed to the final version of the paper.

**Competing interests**

The authors declare that they have no conflict of interest.

**Acknowledgements**

This study is part of the IDEFESE project (https://idefese.wordpress.com/) funded by ADEME, the French Ministry for an Ecological Transition (CGDD and PUCA), and AgroParisTech. We wish to thank the Institut Paris Region for providing important data necessary for the success of the study. PH acknowledges additional funding from the National Research Foundation, Prime Minister's Office, Singapore under award NRF-NRFF12-2020-0009.

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
