# Peer review of "Calibrating and validating the InVEST urban cooling model: Case studies in France and the United States"

_EGUsphere, 2023_

## Author Response (AR1)

**Response to reviewers**

*NB: Line numbers refer to the manuscript version with track changes.*

**Reviewer 1:** **https://doi.org/10.5194/egusphere-2023-928-RC1**

We thank the reviewer for their thorough review of the paper, made possible by their extensive expertise in the topic, which will improve the manuscript. Below we respond to the few comments and suggestions outlined by the reviewer.

- State of the art: "The authors are encouraged to more clearly address the questionable practice to model air temperatures and compare them with LST, as the latter cannot reveal the effect of shading included in the models."

Response: We agree with the reviewer. We have added, l. 312: " We highlight, however, that the model was initially not developed for land surface temperatures. The fair performance for land surface temperatures is an artifact of the model's simplified representation of air temperatures, which imperfectly represents the local energy balance, and the strong correlation of both air and surface temperatures with LULC."

- Paper objectives: "I detected some mismatch between the objectives, outlined in the introduction, and the abstract. The development of a calibration algorithm is mentioned first in the abstract. The reader might expect a detailed explanation of the software tool, its strength and weaknesses. In fact, however, only the calibration and not on the tool itself is reported on."

Response: We thank the reviewer for this comment. We have clarified in the abstract that we "further" developed an existing algorithm (as indeed the improvement is modest compared to the initial development of the algorithm. We have also clarified which improvements were made for this study: " The main improvement on the calibration tool developed for this study is the ability to use as reference temperatures either point data (e.g., a network of stations) or raster data (as is the case for the temperature data in this study). Other improvements are minor and of technical nature (as documented in the source code). The tool reports several performance metrics: mean absolute error (MAE), root mean square error (RMSE), and r2. " (l. 216)

- *Calibration* for the Paris example: "The question arises why averaged ETo (August 1985 – 2005) was chosen as input data for a specific defined heat wave. Wouldn't it have and more appropriate to calculate ETo for mid-August using the FAO ETO-calculator."

Response: We agree that using the FAO ET0-calculator is another option. However, in order to minimize differences in model inputs, we have decided to use the outputs from the ALADIN model that is also used in the reference (modeled) data.

- Calibration for the Twin Cities of Minneapolis/St. Paul: "The readers would like to know more about the density of the network. Was it dense enough to derive 1-km

rasterized temperatures? Was it possible to account for the spatial patterns of the LULC presumably exhibiting different cooling capacities?"

Response: We agree this is an important point to consider the quality of the reference data. We have added the details on the dense network and interpolation from Smoliak et al's data: "(~170 stations over 5000 km2, interpolated with by cokriging using impervious surfaces" (l.205)

- Sensitivity analyses: "This points to the general question if we have enough diversified and differentiated crop coefficients for the calculation of the evapotranspiration from ETo. May I suggest to discuss this issue in the respective section."

Response: We agree and we have made this point explicit in the Discussion of the uncertainties in biophysical values ("(in particular crop coefficients – notably difficult to ascertain for urban land uses)", l. 354)

- "The strong correlation between modeled and observed air temperatures in the Paris example might partially be explained by similarities of the methods used in both data sets. The TEB/Surfex model considered the energy and water budget of Land surfaces. Taking ETo, InVEST also considers the energy balance. I think this is worth to be commented in the paper."

Response: While we agree with the similarities in basic principles, we highlight that the models are very different in their mathematical approaches. TEB/Surfex is much more refined temporally and able to represent more complex physical processes. As such, we decided not to emphasize the similarities in models, but rather that they are both heavily influenced by LULC.

**Reviewer 2: https://doi.org/10.5194/egusphere-2023-928-RC2**

We thank the reviewer for their thorough review of the paper, which clearly shows their expertise in the field. Below, we respond to the specific comments.

**Specific comments**

- Can I assume that the model run before the calibration used the default parameter values as recommended by the InVEST urban cooling model user guide? Maybe that can then made more explicit in the manuscript. In line of that, as Table 2 shows no calibrated values for Paris (for daytime) can we assume that the parameter values have been not changed in the 'after calibration'?
- The author may shortly also explain the performance criteria used (r2 and MAE) (using spatial data, right?)

Response: We have clarified these points in the manuscript:

- (l. 212): "The model outputs were first assessed without calibration, using default parameter values (500 m, 100 m, 0.2, 0.2, 0.6, for $r_{mix}$, $d_{cool}$, $W_a$, $W_e$, $W_s$, respectively)."
- L. 268: "Because of the very low $r^2$ values for daytime temperatures in Paris, we considered calibration unsuccessful and we do not report the calibrated values."

- L. 219: "We selected these metrics since MAE and RMSE are useful quantification of the uncertainty in model outputs, which is important from a user perspective. However, MAE and RMSE also depend on $UHI_{max}$, which means that performance might be artificially good for areas with small urban heat island magnitudes. For this reason, we also report $r^2$ (the default performance criterion for the optimization)."

- InVEST does not predicts LST, but they authors seem to expect a generally good correlation between LST and air temperature (as mentioned in line 70-71; does it account for both, day and night time temperature)? Maybe that could be a bit more discussed in the manuscript, to explain to the reader why the study also included LST in their assessment (and not using reference temperature estimates only).

Response: We agree with this point that was also raised by Reviewer 1. We have added some clarifications in the discussion, in addition to the point in the Methods. L. 312: " We highlight, however, that the model was initially not developed for land surface temperatures. The fair performance for land surface temperatures is an artifact of the model's simplified representation of air temperatures, which imperfectly represents the local energy balance, and the strong correlation of both air and surface temperatures with LULC."

- Line 175 – 177: could the authors add a short description (in the main text or supplementary) of how the different parameter values were assigned, e.g. did the authors use default setting? Or was another method applied? Was it done differently in the two case studies?

Response: As there are no default values for the biophysical parameters, we have derived the values from the literature. The references are provided in Appendix B, with the full biophysical table provided in supplementary data. The reviewer is correct that the methods are different for each case study, since some sources are specific to the case study (e.g., the (APUR, 2020) reference, cited in the main manuscript, is a report with values measured or modelled for Paris). We have also clarified that the $UHI_{max}$ values and $T_{ref}$ temperatures for the Twin Cities case studies were derived differently than for the Paris case study. We highlight this limitation in the Discussion:

l. 328: "The poor performance of the model for daytime air temperature may also be attributed to errors in parameterizations, in particular the use of climate data for short periods (e.g., Aug 6-13$^{th}$ 2003 for Paris) vs. average values over several months, as was the case for some inputs (e.g., reference evapotranspiration in the Paris case study, or $T_{ref}$ and $UHI_{max}$ in the Twin Cities, see Section 2.3). Further investigation of these temporal dynamics should be explored in future work, although we highlight that they did not seem to impact the fair performance of the model for nighttime air temperatures or land surface temperatures."

- Line 199 – 207: different years have been used for the temperature data for the Twin Cities case study, while the authors state to 'study the regional heat wave event in July 22, 2016' (line 123). This seems to be not correct? Similar, I was wondering why the authors used monthly reference evapotranspiration data averaged for the period 1985 – 2005 instead of monthly data for the year 2003 (as done for the temperature data)? Could you explain why?

Response: We thank the reviewer for highlighting these inconsistencies, which we have clarified in the manuscript. For the Twin Cities case study, we have modelled the regional heat wave event for LST but the average summertime temperatures for 2011-2014. (We have clarified this with better

punctuation l. 122) In general though, we agree that climate variables are not consistently derived from climate data series for both case studies. For reference evapotranspiration, the rational is that the variable is only used for its spatial distribution rather than absolute values. We have acknowledged these temporal inconsistencies in the Discussion, highlighting that these were not investigated in this study.

L. 331: "Further investigation of these temporal dynamics should be explored in future work, although we highlight that they did not seem to impact the fair performance of the model for nighttime air temperatures or land surface temperatures."

- Line 257 – 259: could the author shortly elaborate on the the changes of parameter values when compare to values used before the calibration. For example, did the value changed a lot, hence stressing the need to calibrate model parameters instead of using default model settings?

Response: We have added this information to facilitate interpretation for the readers:

l. 270: "The shade weight remains the highest after calibration, and values only changed by <15%."

We have also clarified earlier the default values:

l. 212: "The model outputs were first assessed without calibration, using default parameter values (500 m, 100 m, 0.2, 0.2, 0.6, for $r_{mix}$, $d_{cool}$, $W_a$, $W_e$, $W_s$, respectively)."

- Table 1 shows for the Twin Cities a higher correlation for daytime air temperature with LST after the calibration. Maybe the authors could elaborate a bit more on it (e.g., in line 287)

Response: We agree that the improvement for LST in the case of the Twin Cities is notable. We posit that this could be due to the LULC configuration of the Twin Cities but our analyses do not allow us to ascertain why the calibration was more effective for this dataset. We have highlighted this as an area for future research:

l. 300: "Only in the Twin Cities for surface temperature did the calibration significantly improve model performance, possibly due to the LULC configuration in this landscape, although our analyses do not allow us to confirm this hypothesis"

**Technical corrections**

- Supplementary data of Paris case study: It would be helpful to translate the biophysical table also to English. Also I recommend to add either a short description of how the values have been developed (e.g., which method was used to estimate Kc) or using a reference where the method is further described.

Response: We have translated the biophysical table (supplementary data). The methods to derive biophysical table values was also added to the Appendix (for the Paris case study, while it was already published for the Twin Cities, with the reference provided in the Appendix).

- Line 234, 236, 245-246: suggesting to avoid interpretations like 'overestimating' and 'underestimating' but being descriptive e.g., InVEST model shows higher temperature in forested areas when compared to the reference data; as references data are also obtained from a model rather than obtained from temperature measurements in the field.

Response: We argue that the term "overestimate" and "underestimate" are routinely used in modelling papers and do not infer a personal interpretation, but an objective bias (positive or negative)

- Line 253: I do not understand where the 'confirmation' refers to. Does this refer to the sentence before regarding the very low correlation for daytime air temperature in Paris? Can that be explained with the results of the sensitivity analysis with LST data?

Response: The confirmation refers to the limited effect of the calibration, i.e. if the model is not very sensitive to the parameters, using those parameters as calibration parameters will not have a strong effect. We have clarified this in the next sentence: l. 266: "which could explain the limited effect of calibration".

- Line 214: the link to the user guide does not work

Response: We have fixed the link to the user guide.

---

## Author Response (AR2)

**Response to the Editor**

We thank the Editor for the final review of the manuscript. We respond to the minor comments below. Line numbers refer to the manuscript with track changes.

• *Calibration for the Paris example: "The question arises why averaged ETo (August 1985 – 2005) was chosen as input data for a specific defined heat wave. Wouldn't it have and more appropriate to calculate ETo for mid-August using the FAO ETOcalculator."*

*Response: We agree that using the FAO ET0-calculator is another option. However, in order to minimize differences in model inputs, we have decided to use the outputs from the ALADIN model that is also used in the reference (modeled) data*

*Editor: Could you explain physical implications of this decision to use ETo in this study?*

> Response: We note that using ETo is a requirement for this model (as opposed to actual ET). However, we agree that readers might want more details on the implications of our choice, and we have also added some text to that effect (l. 175): "Using long-term average reference evapotranspiration instead of 2003 reference evapotranspiration has a limited effect on outputs given that the InVEST model only uses relative values (see Eq. 3 above), which have lower temporal variability than absolute values."

• "Other improvements are minor and of technical nature (as documented in the source code)."

>> Please add the information on minor and technical improvements in the main text, not in the source code.

> Response: We have added the details of the improvements (l. 220): "They include testing for the compatibility of user inputs, updating deprecated packages, and improving the code efficiency and readability".

• The author added the sentence of "We selected these metrics since MAE and RMSE are useful quantification of the uncertainty 220 in model outputs, which is important from a user perspective."

>> Could you add references to explain why this is important from a user perspective and what a user perspective implies?

> Response: We have added the following details (l. 222): "Following previous work (Bosch et al., 2021), we selected these metrics since MAE and RMSE are useful quantification of the uncertainty in model outputs with physical quantities (expressed in °C), which is important for users to understand the impact of errors."

• Can you show the distance as km, not latitude/longitude in all maps in the figures (Fig. 1-3)

Response: We thank the editor for the suggestion to improve figures. We note that distances are conveyed by the scale bar and are expressed in km, while the tick marks on the figures refer to the latitudes and longitudes. We have changed them to geographic coordinates (latitude and longitude) for easier reading and following best practice.